# Covalent Triazine Framework C_6_N_6_ as an Electrochemical Sensor for Hydrogen-Containing Industrial Pollutants. A DFT Study

**DOI:** 10.3390/nano13061121

**Published:** 2023-03-21

**Authors:** Hassan H. Hammud, Muhammad Yar, Imene Bayach, Khurshid Ayub

**Affiliations:** 1Department of Chemistry, College of Science, King Faisal University, Al-Ahsa 31982, Saudi Arabia; 2Department of Chemistry, COMSATS University Islamabad, Abbottabad Campus, KPK, Islamabad 22060, Pakistan

**Keywords:** industrial pollutants, covalent triazine framework C_6_N_6_, density functional theory, QTAIM analysis

## Abstract

Industrial pollutants pose a serious threat to ecosystems. Hence, there is a need to search for new efficient sensor materials for the detection of pollutants. In the current study, we explored the electrochemical sensing potential of a C_6_N_6_ sheet for H-containing industrial pollutants (HCN, H_2_S, NH_3_ and PH_3_) through DFT simulations. The adsorption of industrial pollutants over C_6_N_6_ occurs through physisorption, with adsorption energies ranging from −9.36 kcal/mol to −16.46 kcal/mol. The non-covalent interactions of analyte@C_6_N_6_ complexes are quantified by symmetry adapted perturbation theory (SAPT0), quantum theory of atoms in molecules (QTAIM) and non-covalent interaction (NCI) analyses. SAPT0 analyses show that electrostatic and dispersion forces play a dominant role in the stabilization of analytes over C_6_N_6_ sheets. Similarly, NCI and QTAIM analyses also verified the results of SAPT0 and interaction energy analyses. The electronic properties of analyte@C_6_N_6_ complexes are investigated by electron density difference (EDD), natural bond orbital analyses (NBO) and frontier molecular orbital analyses (FMO). Charge is transferred from the C_6_N_6_ sheet to HCN, H_2_S, NH_3_ and PH_3_. The highest exchange of charge is noted for H_2_S (−0.026 e^−^). The results of FMO analyses show that the interaction of all analytes results in changes in the E_H-L_ gap of the C_6_N_6_ sheet. However, the highest decrease in the E_H-L_ gap (2.58 eV) is observed for the NH_3_@C_6_N_6_ complex among all studied analyte@C_6_N_6_ complexes. The orbital density pattern shows that the HOMO density is completely concentrated on NH_3_, while the LUMO density is centred on the C_6_N_6_ surface. Such a type of electronic transition results in a significant change in the E_H-L_ gap. Thus, it is concluded that C_6_N_6_ is highly selective towards NH_3_ compared to the other studied analytes.

## 1. Introduction

In the last few decades, industrial pollutants have become serious threats to living beings. Industrial pollutants are adversely deteriorating our ecosystem. It is reported that every year, 4.2 million people are affected by direct exposure to industrial pollutants at levels beyond the safety limits [1]. Among industrial pollutants, hydrogen-containing substances such as HCN, H_2_S, NH_3_ and PH_3_ are highly poisonous to humans [2,3,4,5]. Hydrogen cyanide (HCN) is a poisonous gas with an almond-like odour. HCN is used as a reagent for the synthesis of various synthetic fibres, pesticides, plastics and dyes [6]. In cases of contact with skin, ingestion and inhalation, HCN is highly lethal to humans. HCN may cause suffocation, unconsciousness, cramps and heart problems. Hydrogen sulphide (H_2_S) is a gas with a rotten egg smell which is usually produced by decomposition of organic materials [7]. It rapidly reacts with the haemoglobin in blood and reduces the oxygen carrying capacity of vital organs. In addition, the detection and removal of H_2_S is particularly essential in petrochemical manufacturing and the coal industry. Another industrially demanding chemical is NH_3_, which is a colourless gas with a pungent smell. It is most widely used in the pharmaceutical and fertilizer industries [8,9]. Phosphine is another hydrogen-containing industrial pollutant, which is used to kill insects. In the semiconductor industry, PH_3_ is applied in silicon processing [10].

The above-mentioned facts demonstrate the need to design an effective electrochemical sensor for environmental monitoring of these hazardous gases. In the recent past, numerous materials such as ZrO_2_ nanoparticles [11], carbon nanotubes [10], metal organic frameworks [11,12], porous silica [13], carbon nitride (C_2_N) [13], hexagonal boron nitride [14], metal oxides [15] and zeolites [16] have been tested as adsorbents and gas sensors [12,17]. In addition, nanocages such as aluminium nitride [18], boron nitride [19], carbon nitride [20], metal clusters [21] and phosphorus carbide [22] have been applied as sensors. The essential pre-requisites for any material to act as a sensor are a high selectivity and sensitivity, a fast response, ease of handling, a low hazard to the environment, a high surface to volume ratio and a high recyclability [23,24]. Materials with a large surface to volume ratio can efficiently interact with analytes. The scientific community has developed new materials to overcome the existing limitations of designed materials, such as a low number of active sites, a lack of a tuneable band gap, a low recyclability, a high cost, a low surface area and selectivity and sensitivity issues. Graphene and graphene-doped materials are among the materials used as sensor materials for toxic and hazardous gases [25,26,27,28,29]. The honeycomb-like structure with a large surface area makes graphene an exceptional material for energy storage [23]; photonics [20]; chemical [22], mechanical [30] and DNA sensing [19]; electrocatalysis [2]; electrochemical sensing [31]; and electronic [19] applications. Similarly, other 2D materials such as germanene [32], transition metal dichalcogenides (TMDs) [33], silicene [34,35,36], phosphorene [37], etc., also offer abundant adsorption sites for separation and sensing of the desired materials. However, the zero band gap of graphene causes a switch off problem in optoelectronic devices [38]. Engineered graphene shows improved electronic charge mobility [39,40]. Many attempts have been made to increase the band gap of graphene through physiochemical engineering. Still, it is not easy to produce a material with a controlled pore size and a suitable band gap [41]. Certainly, the above-mentioned materials have many beneficial characteristics, but their limitations, such as low recyclability, low surface area, low number of active sites and high cost, cannot be ignored. In addition, many surfaces used as sensors contain heavy metals, causing a disposal issue.

Another newly developed class of 2D and 3D materials are covalent organic frameworks, which consist of a variety of organic units as monomers. In 2005, Yaghi et al. synthesized COF1 through a mechanism based on reticular chemistry [42]. Covalent organic frameworks are crystalline porous materials with controlled pore sizes, well-defined surface areas, low densities, light weights and high thermal stabilities. Due to these characteristics, COFs have attracted a lot of interest in various fields of material science, such as solar cells, gas storage, proton conductivity, catalysis, energy storage devices, etc. [43]. In the recent past, several COFs have been utilized as sensors [44] for trinitrotoluene (TNT), picric acid [45], chromium (Cr), arsenic (As) [46,47] and iodine [48]. Previous reports show that covalent triazine frameworks (CTFs) are better sensors compared to COFs due to their controlled porosity, high density nitrogen cavities, high surface areas, high chemical and thermal stabilities, reproducible sensing responses and tuneable electronic properties [49,50]. These characteristics provide suitable conditions for the trapping and detection of analytes and infer the increased potential of CTFs as sensor surfaces compared to COFs. However, CTFs have been studied experimentally as sensors for harmful gases and heavy metals [51,52,53].

Keeping in view the above-mentioned characteristics of covalent triazine frameworks (CTFs), we selected a recently designed covalent triazine framework (C_6_N_6_), which contains six nitrogens in its cavity. Recently, C_6_N_6_ has been used in many fields such as catalysis [54], sensing [55], drug delivery [56] and energy storage [57]. These findings show that the C_6_N_6_ surface is more comparable to other available COFs for the detection of H-containing industrial pollutants such as HCN, H_2_S, NH_3_ and PH_3_. The high amount of nitrogen in the C_6_N_6_ unit also leads to a high surface area and a cavity size of 5.46 Å, which can efficiently interact with various analytes through H bonding, π–π stacking or other types of chemical interactions. According to the best of our knowledge, C_6_N_6_ has not yet been tested as a sensor for the above-mentioned analytes. In the designed study, we used density functional theory simulations to investigate C_6_N_6_ as a sensor surface. Changes in geometrical properties are explored through NCI, QTAIM and SAPT0 analyses, while changes in the electronic properties of analytes in C_6_N_6_ complexes are determined by NBO, EDD and FMO analyses.

## 2. Computational Methodology

Gaussian 09 software was used to carry out the simulations in the current study. The selected geometries of bare C_6_N_6_ and analyte@C_6_N_6_ complexes were investigated at the Ꞷb97XD/6-31G (d, p) level of theory. The Ꞷb97XD functional was adopted in the current study because it is a range-separated functional which can effectively capture non-localized intermolecular interactions [58]. In Ꞷb97XD, the symbol “Ꞷ” represents the long range correction [59]. Additionally, the Ꞷb97XD functional also contains Grimme’s D2 dispersion factor to efficiently study the van der Waals (vdWs) interactions [59]. The changes in the electronic properties of C_6_N_6_ before and after adsorption of analytes were explored by NBO, EDD and FMO analyses at the B3LYP/6-31G(d) level of theory. B3LYP/6-31G(d) is considered as the best to study the electronic changes of interacted systems [60,61,62,63,64,65]. The most stable configurations of the analyte@C_6_N_6_ complexes were investigated by adsorption of analytes through different orientations of C_6_N_6_. A frequency analysis was performed to confirm that the geometries of the complexes were true minima on the potential energy surface. The energies of adsorbed analytes in C_6_N_6_ are determined by Equation (1):∆E = [E_(analytes@C6N6)_ − (E_C6N6_ + E_analytes_)](1)
where E_(analytes@C6N6)_ is the interaction energy of the complexes, E_C6N6_ is the energy for bare C_6_N_6_ and E_analytes_ is the energy of the industrial pollutants used as analytes.

A SAPT0 analysis was applied to determine the role of the individual components of the interaction energy in the stabilization of the analyte@C_6_N_6_ complexes [66]. The total SAPT0 energy is the sum of four factors: induction (∆E_ind_)**,** dispersion (∆E_disp_), electrostatic (∆E_elstat_) and exchange (∆E_exch_). Hence, the SAPT0 energy is calculated through Equation (2):∆E_SAPT0_ = ∆E_elstat_ + ∆E_exch_ + ∆E_ind_ + ∆E_disp_
(2)

The nature of adsorption, such as the electrostatic, dispersive and repulsive forces between the analyte and C_6_N_6_, was examined by an NCI analysis. The nature and strength of non-covalent interactions is given by the reduced density (s), Equation (3) [67].
(3)$$s=\frac{1}{{2(3\pi^{2})}^{\frac{1}{3}}}\frac{\nabla \rho }{\rho^{4/3}}$$

The value of electron density *(ρ)* for non-covalent interactions is generally very small. However, small changes in *(ρ)* result in prominent changes in RDG values, which appear in the form of spikes in a 2D-RGD plot. An NCI analysis characterizes non-covalent interactions through 3D-isosurfaces and 2D-RGD plots. These plots use *ρ* and the second eigen value (λ_2_) to differentiate between the nature of interactions [68,69,70].

The interaction between the analytes and C_6_N_6_ was further studied through a QTAIM analysis. Through a QTAIM analysis, we can examine the nature of the interactions, which cannot be explored through any other method. The major factor through which non-covalent interactions can be studied through QTAIM analysis is the bond critical point (BCP). BCPs are based on various parameters such as the Laplacian ($\nabla^{2}$*ρ*), electron density (*ρ*), total energy density (H), potential energy density (V) and the Lagrangian kinetic energy density (G) [71,72,73]. Multiwfn 3.7 and VMD software were used to perform the QTAIM, NCI and EDD analyses [74,75,76].

## 3. Results and Discussion

### 3.1. Geometry Optimization and Interaction Energy

The optimized structure of the selected model of C_6_N_6_ is presented in Figure 1. The C-C and C-N bond lengths in C_6_N_6_ are 1.53 Å and 1.33 Å, respectively, which are consistent with previously reported theoretical and experimental values [77]. Each unit of C_6_N_6_ comprises of six C_3_N_3_ rings which are connected with each other through C-N bonds [55]. The cavity of the selected model of C_6_N_6_ has a high electron density due to the centring of the nitrogen atoms towards the cavity. The diameter of the C_6_N_6_ cavity (between two nitrogen atoms) is 5.46 Å [78]. The nitrogenated, high electron density cavity of C_6_N_6_ can act as a potential surface for the detection of hydrogen-containing industrial pollutants. The topologies of selected industrial pollutants (HCN, H_2_S, NH_3_ and PH_3_) were also simulated at the Ꞷb97XD/6-31G (d, p) level.

In order to obtain stable complexes of analyte@C_6_N_6_, several orientations of each analyte in C_6_N_6_ were considered. Figure 1 shows a graphical representation of the adsorption energies of analyte@C_6_N_6_ complexes. The most stable geometries of each analyte@C_6_N_6_ complex are reported in Figure 2, while the remaining geometries are given in Appendix A. The following order of the adsorption energies of the stable complexes is obtained: HCN@C_6_N_6_ > H_2_S@C_6_N_6_ > NH_3_@C_6_N_6_ > PH_3_@C_6_N_6_.

Among the analyte@C_6_N_6_ complexes, the most stable complex is formed between HCN and C_6_N_6_, with an adsorption energy value of −16.46 kcal/mol. In this complex, HCN is projected to be perpendicular to the C_6_N_6_ cavity (see Figure 2). The H atom of HCN interacts with the N atoms of C_6_N_6_ at bond distances of between 2.57 Å and 3.09 Å. Among these interactions, the strongest interaction is seen between the H7 atom of HCN and the N1 and N2 of C_6_N_6_, with interaction distances of 2.57 Å and 2.67 Å, respectively. These interaction distances show that HCN is more attracted to the two C_3_N_3_ rings containing N1 and N2 atoms. When the input geometry was altered to have the nitrogen atom of HCN pointing towards the cavity of C_6_N_6_, the calculation converged to the same complex mentioned above (hydrogen pointing towards the cavity). This happened because of the repulsion between the N atom of HCN and the N atoms of C_6_N_6_.

The stable complex of H_2_S@C_6_N_6_ is obtained with an adsorption energy of −13.64 kcal/mol, which is the 2nd most stable complex among all studied analyte@C_6_N_6_ complexes. The optimized topology of the H_2_S@C_6_N_6_ complex shows that both H atoms of H_2_S are positioned towards the C_6_N_6_ cavity. Interaction distances of 2.90 Å (N1---H7), 2.29 Å (N2---H7) and 2.68 Å N2---H7 are noted between N atoms (N1, N2 and N3) of C_6_N_6_ and the H atom (H7) of H_2_S (see Figure 2). Moreover, similar interaction distances are observed between N4---H8, N5---H8 and N6---H8, respectively. However, these interaction distances between the N atoms of C_6_N_6_ and the H atoms of H_2_S reveal that the adsorption of H_2_S occurred at the centre of cavity, while the S atom is pointing upwards.

The stable complex of NH_3_@C_6_N_6_ resulted in an adsorption energy of −12.27 kcal/mol, which is slightly lower than that of the H_2_S@C_6_N_6_ complex. In the stable configuration of NH_3_@C_6_N_6_, two H atoms of NH_3_ interact with the N atoms of C_6_N_6_, while the 3^rd^ H atom is projected away from the cavity. Among the observed interactions, strong interactions are observed between N1 and H5 and N4 and H6 of C_6_N_6_ and H_2_S, with distances of 2.33 Å and 2.34 Å, respectively. The N2 and N3 of C_6_N_6_ interact with H5 and H6 of NH_3_, with an interaction distance of 2.57 Å. The lower adsorption energy of the NH_3_@C_6_N_6_ complex (−12.27 kcal/mol) compared to the H_2_S@C_6_N_6_ complex (−13.64 kcal/mol) is attributed to longer interaction distances between the H atoms of NH_3_ and the N atoms of C_6_N_6_ compared to H_2_S.

An adsorption energy of −9.36 kcal/mol is observed for the stable complex of PH_3_@C_6_N_6_, which is the lowest amongst all studied analyte@C_6_N_6_ complexes. The stable structure of the PH_3_@C_6_N_6_ complex consists of an inward pointing H atom of PH_3_, while the other two H atoms point upwards. The inward pointing H atom of PH_3_ is almost at the centre of the C_6_N_6_ cavity. The bond distances between the H atoms of PH_3_ and the N atoms of C_6_N_6_ range from 2.70 Å to 3.09 Å. The low adsorption energy (−9.36 kcal/mol) of the PH_3_@C_6_N_6_ complex is due to the low polarity of the H atoms of PH_3_ and the longer interaction distances. The results of adsorption energies (−9.36 kcal/mol to −16.46 kcal/mol) show all analytes (HCN, H_2_S, NH_3_ and PH_3_) physically absorb on the C_6_N_6_ sheet.

### 3.2. Symmetry Adapted Perturbation Theory (SAPT0) Analysis

A SAPT0 analysis is the most effective method for quantifying non-covalent interactions. The SAPT0 interaction energy is the sum of four factors, including induction, exchange, electrostatic and dispersion. Therefore, in SAPT0, the role of each factor is investigated in the stabilization of analytes in C_6_N_6_. The SAPT0 results of all the studied complexes are presented in Table 1 and Figure 3. In the SAPT0 graph, five bars are shown for each complex. The last green bar represents the total SAPT0, which is the sum of the first four bars. Out of these four bars, three bars with negative signs indicate attractive interactions, while the bars with +ve signs indicate repulsive interactions. The negative sign of the green bar (total SAPT0) reveals the dominance of attractive forces over repulsive forces during the interaction of analytes with the C_6_N_6_ sheet.

The SAPT0 energies of all the complexes obtained through PSI4 are −16.88 kcal/mol (HCN@C_6_N_6_), −12.26 kcal/mol (H_2_S@C_6_N_6_), −10.06 kcal/mol (NH_3_@C_6_N_6_) and −9.22 kcal/mol (PH_3_@C_6_N_6_). Among the studied H-containing industrial pollutants over C_6_N_6_, the highest SAPT0 value (−16.88 kcal/mol) is obtained for the HCN@C_6_N_6_ complex, which is consistent with the results of the interaction energy analysis. The SAPT0 attractive component values for the HCN@C_6_N_6_ complex are E_elst_ (−14.60 kcal/mol: 52%), E_ind_ (−3.78 kcal/mol: 13%) and E_disp_ (−9.66 kcal/mol: 34%). These values indicate that E_elst_ is the major stabilizing factor compared to E_ind_ and E_disp_. The highest value of E_elst_ results from the strong interactions of the H atom of HCN with the N atoms of C_6_N_6_ through a shorter interaction distance (vide supra).

The second highest SAPT0 value is seen for the H_2_S@C_6_N_6_ complex, at −12.26 kcal/mol. The energy values of the attractive components, i.e., E_elst_, E_ind_ and E_disp_, observed for the H_2_S@C_6_N_6_ complex are −12.58 kcal/mol (43%), −4.27 kcal/mol (15%) and −12.34 kcal/mol (42%), respectively. These values of the attractive components show that E_elst_ (43%) and E_disp_ (42%) play a major role in the stabilization of the H_2_S@C_6_N_6_ complex, while a lower contribution is observed for E_ind_ (15%). The SAPT0 values for the NH_3_@C_6_N_6_ and PH_3_@C_6_N_6_ complexes are −10.06 kcal/mol and −9.22 kcal/mol, respectively. The SAPT0 component analysis for the NH_3_@C_6_N_6_ complex reveals that E_elst_ (47%) and E_disp_ (40%) are dominant in the stabilization of NH_3_ over C_6_N_6_. Among the studied pollutant complexes, the lowest SAPT0 value is observed for the PH_3_@C_6_N_6_ complex (−9.22 kcal/mol). The lowest SAPT0 value for the PH_3_@C_6_N_6_ complex results from the lower polarity of the interacting H atoms of PH_3_ with the N atoms of C_6_N_6_. Among the SAPT0 energy, a key role is noted for E_disp_ (55%) in the stabilization of the PH_3_@C_6_N_6_ complex, while E_elst_ (33%) and E_ind_ (12%) contribute less towards the total SAPT0 energy.

The SAPT0 analysis reveals that E_elst_ is the dominant contributing factor in the case of HCN@C_6_N_6_, whereas a good balance between E_elst_ and E_disp_ is observed for H_2_S and NH_3_. In the case of PH_3_, the dispersion interaction played a dominant role in the stabilization of PH_3_ over C_6_N_6_. The SAPT0 values of all the studied complexes are consistent with the interaction energy analysis.

### 3.3. Non-Covalent Interaction (NCI) Analysis

The nature of interactions between analytes and C_6_N_6_ was further explored through non-covalent interaction analyses. Through an NCI analysis, the van der Waals, electrostatic and repulsive interactions can be differentiated based on a colour scheme. An NCI analysis comprises 3D isosurfaces and 2D reduced density gradient (RDG) plots. The 3D isosurface topological analysis is based on three types of colours: red (repulsive), green (van der Waals) and blue (electrostatic interactions). These colours appear in the form of patches between the analytes and C_6_N_6_ through which the nature of interactions can be differentiated. A 2D RDG plot is obtained by taking the reduced density on the *Y*-axis and product of 2^nd^ value of the Laplacian (sign(λ_2_)) and density gradient on the *X*-axis. A 2D RGD plot gives information about the strength of each type of interaction. In 2D RDG plots, non-covalent interactions appear as blue, green and red spikes in the low density gradient region. The vdWs interactions are indicated by the green spikes which form the when product of sign(λ_2_)ρ is in the range of −0.00 a.u. to −0.02 a.u. A large and negative value of the product of sign(λ_2_)ρ (above −0.02 a.u.) signifies the existence of electrostatic interactions (blue spikes). Red spikes in the 2D-RGD plot show repulsive interactions when the product of sign(λ_2_)ρ is positive and large.

The topologies obtained through NCI analyses are presented in Figure 4. In the studied complexes, the green spikes in the 2D RDG plots and the green isosurface in the 3D plots are explored. The green spikes and isosurface indicate vdWs interactions between the analytes and C_6_N_6_. The strength of the vdWs interaction in each complex is different, indicated by the variation in the thickness of 3D isosurface patches and a projection of green spikes in the 2D-RDG plots on the *X*-axis. In the case of the HCN@C_6_N_6_ complex, a ring on the green isosurface develops between the H atom of HCN and the six N atoms of C_6_N_6_. This shows that all six N atoms of C_6_N_6_ strongly interact with the H atom of HCN. Similarly, in the 2D-RDG plot, a mixture of bluish-green spikes arises at a low-density gradient between −0.01 and −0.02 a.u., which indicates that the complex is stabilized through electrostatic interactions. In the case of the H_2_S@C_6_N_6_ complex, a stronger interaction is observed, where the H atoms of H_2_S closely interact with the N atoms of C_6_N_6_ (see Figure 4). The rest of the interactions are present at longer distances due to the angular orientation of the H_2_S molecule over C_6_N_6_. The 3D isosurfaces and 2D-RGD plots of the NH_3_@C_6_N_6_ complex reveal the weak interaction compared to the HCN@C_6_N_6_ and H_2_S@C_6_N_6_ complexes. The scattered green spikes and shattered green isosurface in the 3D NCI plots are due to the weak interaction between NH_3_ and the C_6_N_6_ cavity. The weak interaction of NH_3_ is also described in the interaction energy section. In the case of the PH_3_@C_6_N_6_ complex, the green spikes and isosurface are not as thick and dense as projected in the rest of the studied complexes. Out of the three hydrogens atoms of PH_3_, only one H is oriented towards the C_6_N_6_ cavity. The H atom orientated towards the C_6_N_6_ cavity is not as polar as the H atom of HCN and NH_3_, due to which weak vdWs interactions are established between PH_3_ and the C_6_N_6_ cavity. The NCI analysis indicated that dispersion and electrostatic interactions are dominant, which is consistent with the SAPT0 and interaction energy results.

### 3.4. Quantum Theory of Atoms in Molecule (QTAIM) Analysis

QTAIM analysis is a useful tool to investigate all non-covalent interactions which are impossible to capture through any other analysis. Through QTAIM analysis, inter- and intra-molecular interactions such as ionic interactions, hydrogen bonding, van der Waals interactions and covalent bonds can be studied. These inter- and intra-molecular interactions are studied by various parameters such as cage critical points (CCP), nuclear critical points (NCP), ring critical points (NCP) and bond critical points (BCPs). In QTAIM analyses, non-covalent interactions are best explained by BCPs. BCPs between two interacting systems are represented by a brown line (see Figure 5). The nature and strength of each BCP is explained through several parameters such as ($\nabla^{2}$*ρ*), (ρ), (H), (G) and (V). The values of the BCP parameters of all complexes are given in Table 2.

The nature of interaction is shared if $\nabla^{2}$*ρ*(r) < 0, whereas for close shell interactions, the value of $\nabla^{2}$*ρ*(r) > 0. For electrostatic interactions (H bonding), the values of ($\nabla^{2}$*ρ*) and (ρ) at BCPs are between 0.024 and 0.139 a.u. and 0.002 to 0.034 a.u., respectively [79,80]. In addition, a bond distance (N---H, O---H and F---H) of less than 1.2 Å indicates a strong interaction, while a bond distance greater than 1.8 Å shows a weak interaction [81]. The strength of an individual bond can also be characterized by the Espinosa approach, as presented in Equation (4) [82]. For H bonding, the energy value of an individual bond is >3 kcal/mol (in negative) [83].
(4)$$E_{\mathrm{int}}\;(a.u.)=\frac{1}{2}V(r)$$

In addition, the BCP can further be investigated by Equation (5):H(r) = G(r) + *V*(*r*)(5)
where G, H and V are the kinetic, potential and total energy density at the BCPs, respectively. For close shell interactions, the value of H is usually positive, while it is negative for shared shell interactions. The type of bonding is generally indicated by $\nabla^{2}$*ρ* and H. The values of $\nabla^{2}$*ρ* and H are less than zero for shared shell interactions and greater than zero for close shell interactions. If the values of $\nabla^{2}$*ρ* are greater than zero and H is less than zero, it indicates H bonding. Another parameter used to explain the BCP is the ratio -V/G. If the ratio of -V/G is >2, it indicates covalent bonding, and a ratio of −V/G of < 1 represents non-covalent interactions [84,85].

QTAIM analyses of all the complexes shows that number of BCPs is six in each case due to interactions of six N atoms in the C_6_N_6_ cavity with the H atoms of interacting analytes. However, the strengths and types of bonding in all complexes are different due to the different values of their BCP parameters (see Table 2). In the case of the HCN@C_6_N_6_ complex, the $\nabla^{2}$*ρ* and *ρ* values are in the range of 0.02429 a.u. to 0.0284 a.u. and 0.00706 a.u. to 0.00814 a.u., respectively. The most stable interaction is observed between H7 of HCN and N6 of C_6_N_6_. The values of $\nabla^{2}$*ρ* and *ρ* at BCPs between H7 and N6 are 0.0284 a.u. and 0.00814 a.u., respectively. In addition, the rest of the BCP parameter values of the HCN@C_6_N_6_ complex are in the range of electrostatic attractions, which is consistent with the SAPT0 analysis, where the dominant contribution was electrostatic. The other BCP parameters such as G, H, V, -V/G and E_int_ are in the range of electrostatic interactions.

A topological analysis of the H_2_S@C_6_N_6_ complex also shows six BCPs. Among these BCPs, two BCPs (H7---N2 and H8---N5) are in the strong electrostatic region, while the rest of the BCPs indicate dispersion interactions (see Table 2). Again, the results of the QTAIM analysis for the H_2_S@C_6_N_6_ complex are well matched with the SAPT0, where a good balance between electrostatic and dispersion interactions was observed. The highest values of $\nabla^{2}$*ρ* (0.03965)*, ρ* (.01586 a.u)*,* H (0.00 a.u.) and E_int_ (−3.11 kcal/mol) are observed for H7---N2 and H8---N5, respectively. The BCP parameter such as $\nabla^{2}$*ρ, ρ* and E_int_ of H7---N2 and H8---N5 are in the range of H bonding, but no such types of blue isosurface and spikes are seen in the 3D isosurfaces and 2D-RGD plots of NCI analysis. The interaction distances between H7 and N2 and H8 and N5 are 2.29 Å each, which is greater than the 1.8 Å required for interacting system to demonstrate H bonding [81].

The BCP parameter values of the NH_3_@C_6_N_6_ complex are in the range of medium to strong interactions between NH_3_ and C_6_N_6_. The strongest interactions are observed between H7 and N2 and H9 and N6 of NH_3_ and C_6_N_6_, respectively. The values of these two BCP parameters $\nabla^{2}$*ρ, ρ* and E_int_ are almost comparable (see Table 2). The values of the rest of the five BCPs indicate vdWs interactions. The number of BCPs for the PH_3_@C_6_N_6_ complex are equal in number with other complexes. However, all BCP parameter values between the H atoms of PH_3_ and the N atoms of C_6_N_6_ are in the range of dispersion interactions, which is consistent with the SAPT0 interaction energy analysis, where PH_3_@C_6_N_6_ was the least stable complex. In all studied analyte@C_6_N_6_ complexes, the values of BCP parameters are in good agreement with NCI and SAPT0 analyses.

## 4. Electronic Properties

### 4.1. Natural Bond Orbital (NBO) and Electron Density Differences (EDD) Analyses

The electronic properties of complexes were explored by EDD and NBO analyses. Through these analyses of the nature of interactions, the stability, charge transfer and sensing ability of the designed surface for selected analytes can be explored. An EDD plot is obtained by subtracting the charges of individual fragments from the charge of the analyte@C_6_N_6_ complex. The topologies obtained through EDD analyses show two types of isosurface: yellow and green. The yellow surfaces reveal a depletion in charge, while a green colour illustrates an accumulation of charge. Among the studied complexes, both types of isosurface are observed in the adsorption site of analytes and the C_6_N_6_ surface, which indicates an exchange of electronic charge between them. The topologies of the complexes determined through EDD analyses are presented in Figure 6. In addition, the NBO charge values of each analyte over C_6_N_6_ are also displayed in Figure 6.

The charge transfer values from C_6_N_6_ to the analytes are −0.019 e^−^ (HCN), −0.026 e^−^ (H_2_S), −0.015 e^−^ (NH_3_) and −0.008 e^−^ (PH_3_). These values of charges show that the analytes extracted charge from C_6_N_6_. In all studied analytes, the H atoms of analytes interacted with the electron rich cavity of C_6_N_6_, due to which charge is transferred from C_6_N_6_ to the analytes. The highest charge value is extracted by H_2_S (−0.026 e^−^) due to a close interaction of the H atoms of H_2_S with the N atoms of C_6_N_6_. Similarly, the accumulation and depletion of electron density in the interacting sites of C_6_N_6_ and H_2_S clearly indicate that the nitrogen of C_6_N_6_ transferred more charge towards H_2_S. In the case of HCN, a charge of −0.019 e^−^ is determined, which is contrary to its interaction energies. The EDD analysis of HCN adsorption over C_6_N_6_ shows that charge is transferred from the N atoms of C_6_N_6_ to the H atom of HCN. However, the lowest values of charge are observed in NH_3_ (−0.015 e^−^) and PH_3_ (−0.008 e^−^) due to the weak interactions of these analytes with the C_6_N_6_ sheet.

### 4.2. Frontier Molecular Orbital (FMO) Analysis

The electrical signal produced by an electrochemical sensor mainly depends on changes in its electronic properties after interaction with an analyte. To explore these changes in electronic properties of sensors, a frontier molecular orbital analysis was performed. The energy values determined from an FMO analysis of bare C_6_N_6_ and analyte@C_6_N_6_ complexes are shown in Table 3. The densities of HOMO and LUMO orbitals are presented in Figure 7. The energies of the HOMO and LUMO for C_6_N_6_ are −7.16 eV and −3.30 eV, respectively, with a HOMO–LUMO energy gap (E_H-L_ gap) of 3.86 eV. The interaction of selected industrial pollutants with C_6_N_6_ causes prominent changes in the energy levels of the HOMO, LUMO and E_H-L_ gap. The computed values of the E_H-L_ gaps of HCN@C_6_N_6_, H_2_S@C_6_N_6_, NH_3_@C_6_N_6_ and PH_3_@C_6_N_6_ complexes are 4.02 eV, 2.76 eV, 2.58 eV and 3.35 eV, respectively.

The results presented in Table 3 show that adsorption of H_2_S, NH_3_ and PH_3_ on C_6_N_6_ causes a pronounced increase in the HOMO energy, while the LUMO energy of these complexes remains almost unchanged. However, adsorption of HCN shows the inverse behaviour, where a significant decrease in the HOMO energy (−7.43 eV) is noticed compared to C_6_N_6_ (−7.16 eV). A more pronounced decrease in the HOMO energy causes an increase in the E_H-L_ gap (4.02 eV) of the HCN@C_6_N_6_ complex compared to bare C_6_N_6_ (3.86 eV). This increase in the E_H-L_ gap of the HCN@C_6_N_6_ complex indicates a decrease in the conductivity of C_6_N_6_. As a result, C_6_N_6_ cannot act as a good sensor for HCN.

Adsorption of H_2_S onto C_6_N_6_ resulted in a prominent increase in the energy of the HOMO (−6.07 eV), while the LUMO energy (−3.31 eV) is almost comparable to bare C_6_N_6_ (−3.30 eV). This increase in the HOMO energy resulted in a pronounced reduction in the E_H-L_ gap (2.76 eV) of the H_2_S@C_6_N_6_ complex. Among the different industrial pollutants adsorbed on C_6_N_6_, the most significant increase in electrical conductivity is observed for the NH_3_@C_6_N_6_ complex due to an appreciable decrease in the E_H-L_ gap. The energies of the HOMO and LUMO for the NH_3_@C_6_N_6_ complex are 5.85 eV and −3.27 eV, respectively. The interaction of PH_3_ changes the HOMO (−6.62 eV) and LUMO (−3.27 eV) energies in such a way that a negligible decrease in the E_H-L_ gap (3.35 eV) is observed for the PH_3_@C_6_N_6_ complex. Among the studied analyte@C_6_N_6_ complexes, the highest decrease in the E_H-L_ gap is observed for NH_3_@C_6_N_6_. This indicates that C_6_N_6_ can act as potential surface for this analyte. Thus, it is concluded that C_6_N_6_ is highly selective towards NH_3_ compared to the other studied analytes.

After adsorption of industrial pollutants (HCN, H_2_S, NH_3_ and PH_3_) onto C_6_N_6_, the dispersal of orbitals densities was also explored in order to comprehend the sensing abilities of C_6_N_6_. We observed three types of orbital distribution patterns. In the case of the HCN@C_6_N_6_ complex, the orbital densities are localized on the triazine ring. In the HOMO, the orbital densities are observed on the carbon and nitrogen of the triazine ring, while in the LUMO, the orbital densities are shifted to the carbon of the triazine ring. This shows that analytes do not participate in the electronic shift from the HOMO to the LUMO (see Figure 7).

The intra shift of electronic orbital densities causes an increase in the E_H-L_ gap compared to bare C_6_N_6_. In H_2_S@C_6_N_6_ and PH_3_@C_6_N_6_ complexes, the major portion of the HOMO density is located on the analyte, with some sharing with the N atoms of C_6_N_6_, while the LUMO density is completely shifted onto C_6_N_6_. Such a type of electronic transition causes a significant change in the E_H-L_ gap. A third type of electronic transition is observed in the NH_3_@C_6_N_6_ complex, where the HOMO is completely located on NH_3_, while the LUMO is shifted to C_6_N_6_ (see Figure 7). Such a type of electronic transition brings about a significant change in the E_H-L_ gap. The same sort of behaviour was observed by us previously in a study of the same analytes over C_2_N. The transition of orbital density from analytes to the C_2_N surface causes considerable changes in the E_H-L_ gaps [34,71,72].

We compared the adsorption energies of our studied system with those available in the literature for different COFs. The adsorption energies of H-containing analytes are comparable or somewhat higher than already reported values on other surfaces. An adsorption energy of −12.27 kcal/mol is observed for the NH_3_@C_6_N_6_ complex, whereas in the literature, adsorption energies of −5.27 kcal/mol, −10.68 kcal/mol and −6.65 kcal/mol are observed for NH_3_@CTF-FUM, NH_3_@C_2_N and NH_3_@CTF-0 complexes, respectively. In addition, interaction energies of −15.24 kcal/mol, −3.79 kcal/mol and −2.34 kcal/mol are observed for HCN@CTF-FUM, H_2_S@CTF-FUM and PH_3_@CTF-FUM, respectively. The adsorption energies of the rest of the complexes are given in Table 4 and reveal that C_6_N_6_ can act as a better surface for electrochemical sensing of H-containing analytes.

## 5. Conclusions

The electrochemical sensing application of C_6_N_6_ is evaluated by DFT simulations at the wb97xd/6-31g(d,p) level of theory. The adsorption of industrial pollutants over C_6_N_6_ occurred through physisorption, with adsorption energies ranging from −9.36 kcal/mol to −16.46 kcal/mol. Among the considered analytes, the maximum energy is observed for HCN@C_6_N_6_ (−16.46 kcal/mol). The non-covalent interactions of analyte@C_6_N_6_ complexes are explored through symmetry-adapted perturbation theory (SAPT0), quantum theory of atoms in molecules (QTAIM) analyses and non-covalent interaction (NCI) analyses. The attractive components in SAPT0 analyses show that electrostatic and dispersive interactions play a dominant role in the stabilization of analytes in the C_6_N_6_ cavity. Similarly, NCI and QTAIM analyses also verified the findings of SAPT0 and interaction energy analyses. In addition, the electron density (ρ_BCP_) and Laplacian of electron density (∇^2^ρ_bcp_) remained high for the HCN@C_6_N_6_ complex, which is consistent with the NCI and SAPT0 results. The electronic properties of analyte@C_6_N_6_ complexes are investigated by electron density difference (EDD), natural bond orbital analyses (NBO) and frontier molecular orbital analyses (FMO). Charge is transferred from the C_6_N_6_ sheet to HCN, H_2_S, NH_3_ and PH_3_. The highest exchange of charge is noted for H_2_S (−0.026 e^−^). The results of FMO analyses show that the interaction of all analytes results in changes in the E_H-L_ gap of the C_6_N_6_ sheet. However, the highest decrease in the E_H-L_ gap (2.58 eV) is observed for the NH_3_@C_6_N_6_ complex compared to other studied analyte@C_6_N_6_ complexes. The orbital density pattern shows that the HOMO density is completely concentrated on NH_3_, while the LUMO density is transferred towards C_6_N_6_. This indicates that C_6_N_6_ can act as potential surface for these analytes. Hence, which analyte is more selective towards C_6_N_6_ depends on its electrical conductivity and not its adsorption. Thus, it is concluded that C_6_N_6_ is highly selective towards NH_3_ compared to the other studied analytes.

## Figures and Tables

**Figure 1 nanomaterials-13-01121-f001:**
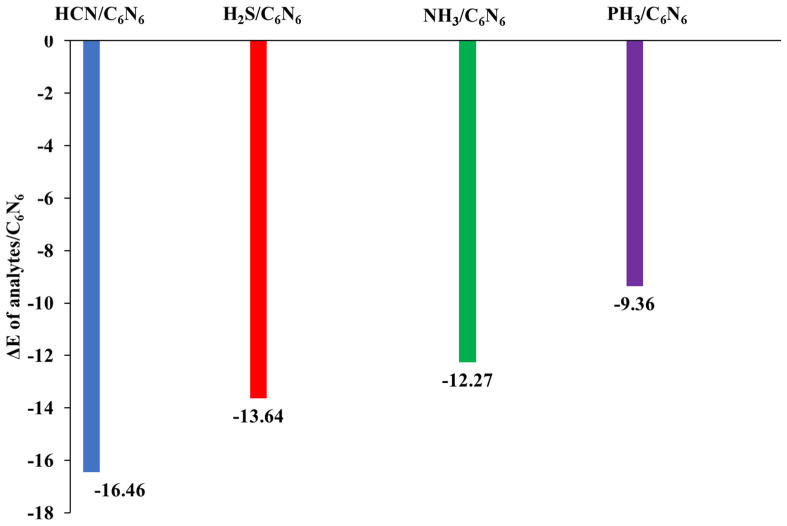
Graphical representation of adsorption energies of analyte@C_6_N_6_ complexes at the Ꞷb97XD/6-31G (d,p) level of theory.

**Figure 2 nanomaterials-13-01121-f002:**
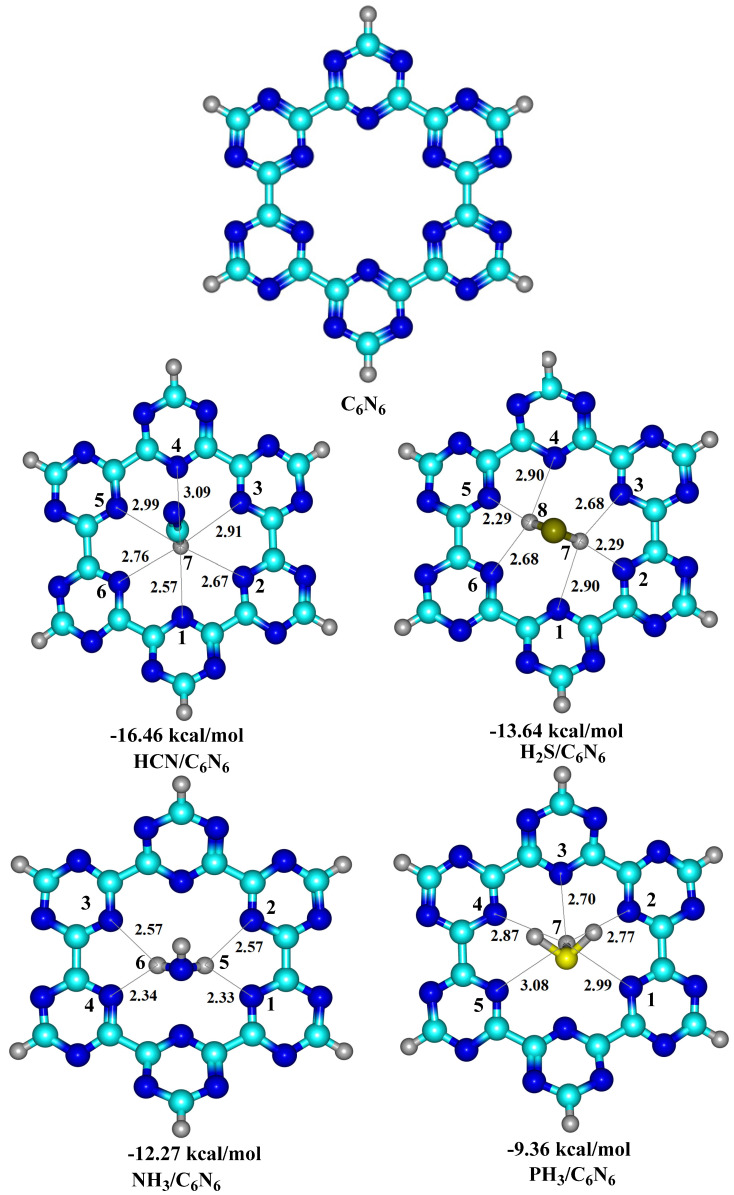
Top view of optimized structures of the analyte@C_6_N_6_ complexes at the Ꞷb97XD/6-31G (d,p) level of theory.

**Figure 3 nanomaterials-13-01121-f003:**
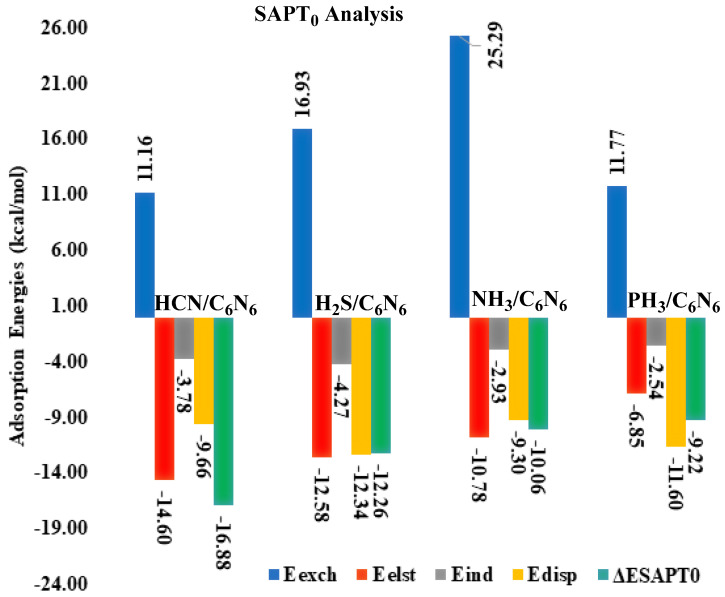
Graphical projection of the SAPT0 energy and its components for analyte@C_6_N_6_ complexes.

**Figure 4 nanomaterials-13-01121-f004:**
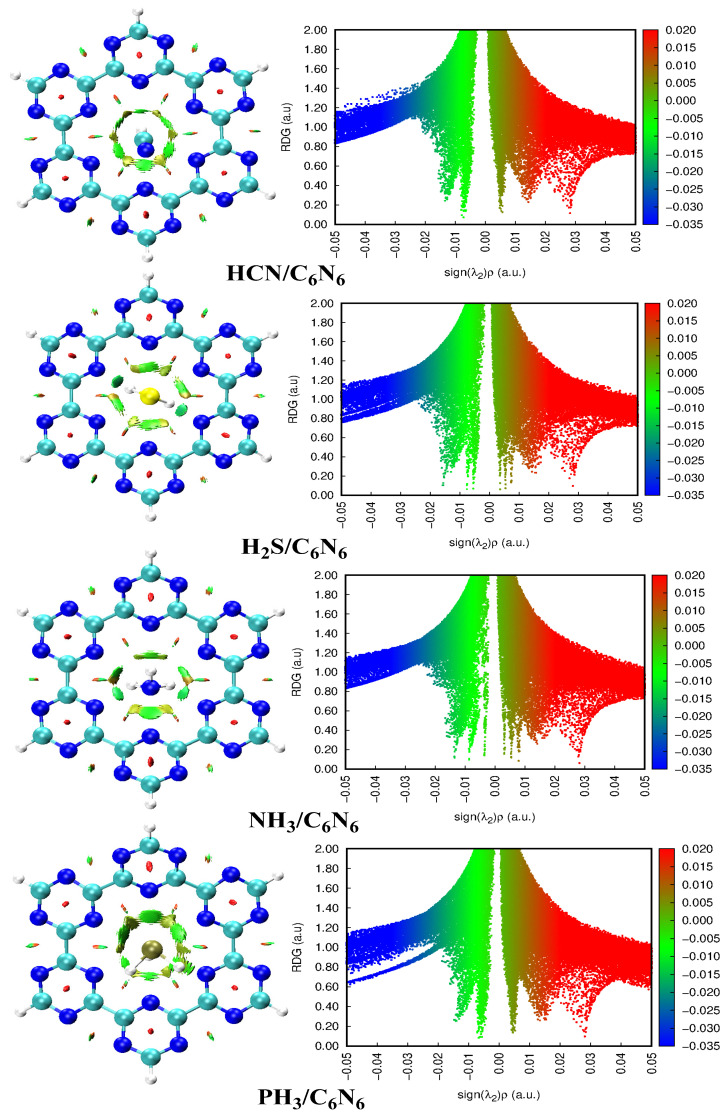
NCI analyses of analyte@C_6_N_6_ complexes at an iso-value of 0.002 a.u.

**Figure 5 nanomaterials-13-01121-f005:**
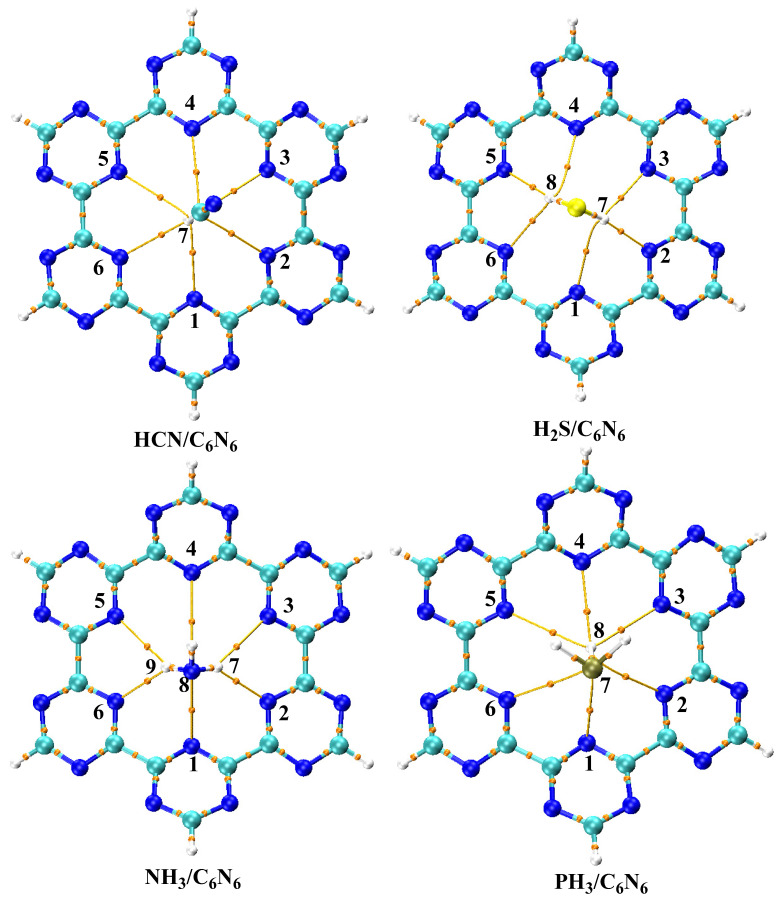
Topological analysis of analyte@C_6_N_6_ complexes obtained through QTAIM analysis.

**Figure 6 nanomaterials-13-01121-f006:**
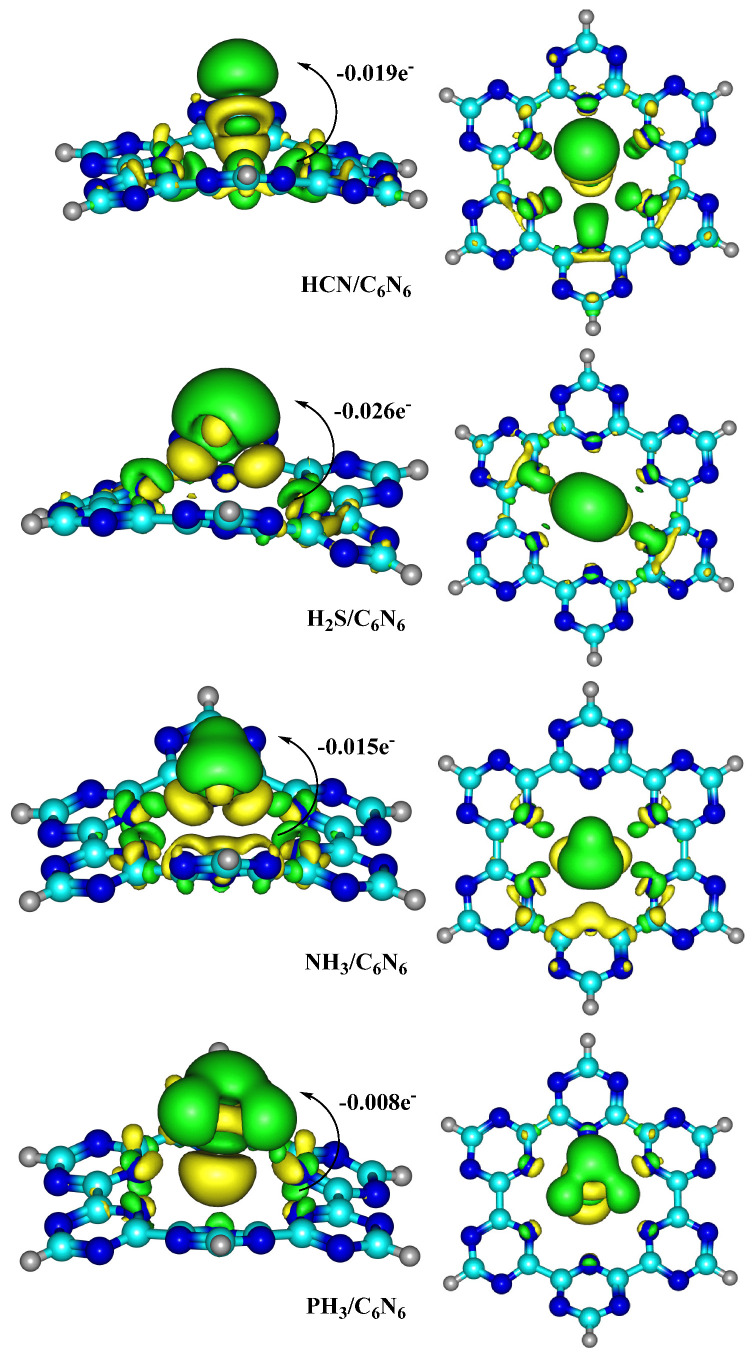
The topologies of analyte@C_6_N_6_ complexes determined through EDD analysis (Isovalue = 0.0019 a.u.).

**Figure 7 nanomaterials-13-01121-f007:**
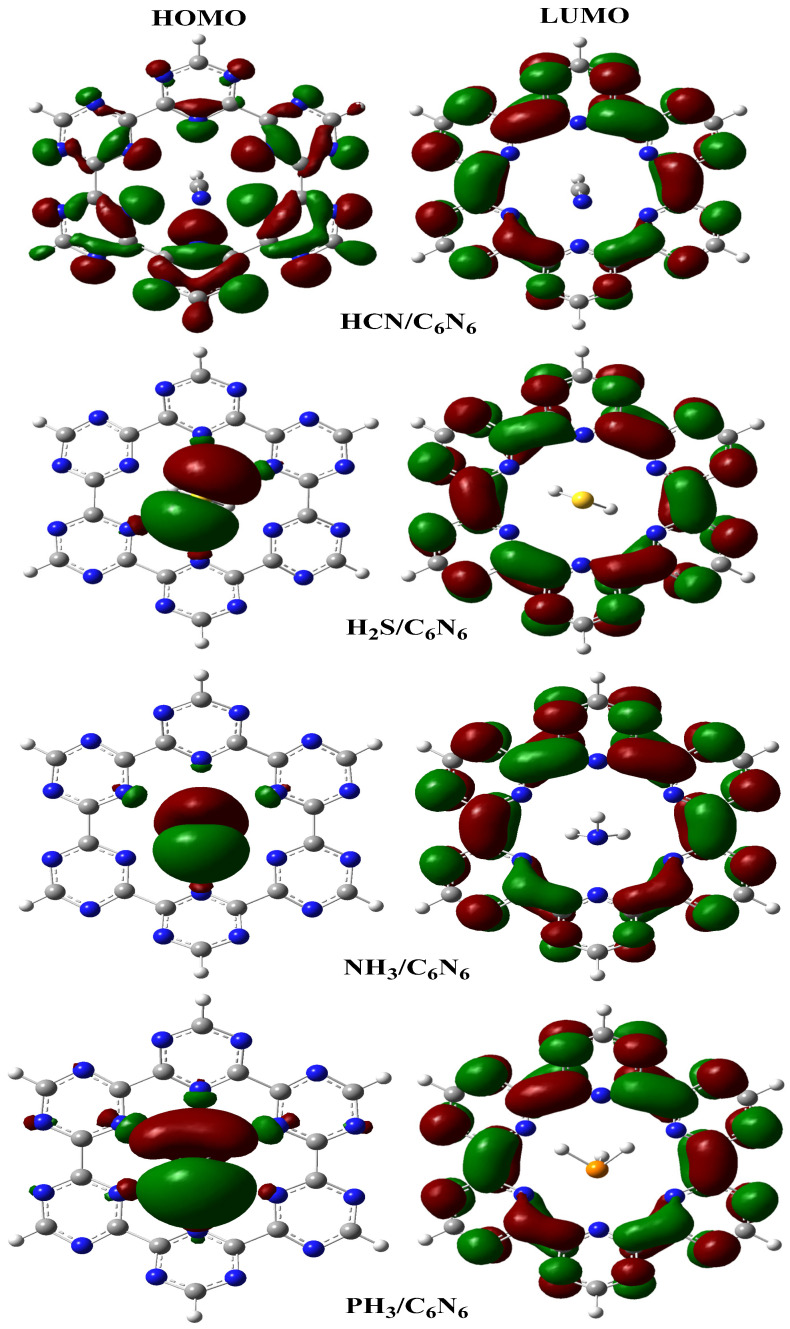
Orbital distribution pattern of analyte@C_6_N_6_ complexes computed through a FMO analysis (isovalue = 0.02 a.u.). Green orbital densities indicate a positive wavefunction, while red orbital densities show a negative wavefunction.

**Table 1 nanomaterials-13-01121-t001:** Stabilization energies of analyte@C_6_N_6_ complexes obtained through SAPT0 analyses.

Analytes@C_6_N_6_	E_exch_	E_elst_	(%)	E_ind_	(%)	E_disp_	(%)	E_SAPT0_
**HCN@C_6_N_6_**	11.16	−14.60	(52)	−3.78	(13)	−9.66	(34)	−16.88
**H_2_S@C_6_N_6_**	16.93	−12.58	(43)	−4.27	(15)	−12.34	(42)	−12.26
**NH_3_@C_6_N_6_**	25.29	−10.78	(47)	−2.93	(13)	−9.30	(40)	−10.06
**PH_3_@C_6_N_6_**	11.77	−6.85	(33)	−2.54	(12)	−11.60	(55)	−9.22

**Table 2 nanomaterials-13-01121-t002:** BCP parameters values of analyte@C_6_N_6_ complexes determined by QTAIM analyses.

Analyte@C_6_N_6_	Analyte-C_6_N_6_	ρ	∇^2^ρ	G (r)	V (r)	H (r)	V(r)/G(r)	Eint (kcal/mol)
**HCN@C_6_N_6_**	H7-N1	0.00724	0.02687	0.00540	−0.00408	0.00132	−0.76	−1.28
	H7-N2	0.00706	0.02429	0.00494	−0.00381	0.00113	−0.77	−1.19
	H7-N3	0.00780	0.02622	0.00547	−0.00439	0.00108	−0.80	−1.38
	H7-N4	0.00735	0.02451	0.00508	−0.00404	0.00104	−0.79	−1.27
	H7-N5	0.00659	0.02509	0.00489	−0.00352	0.00138	−0.72	−1.10
	H7-N6	0.00814	0.02847	0.00597	−0.00483	0.00115	−0.81	−1.51
**H_2_S@ C_6_N_6_**	H7-N1	0.00549	0.01895	0.00380	−0.00285	0.00000	−0.75	−0.90
	H7-N2	0.01586	0.03965	0.00992	−0.00992	0.00000	−1.00	−3.11
	H8-N3	0.00763	0.02552	0.00536	−0.00434	0.00102	−0.81	−1.36
	H8-N4	0.00549	0.01895	0.00379	−0.00285	0.00094	−0.75	−0.89
	H8-N5	0.01585	0.03963	0.00991	−0.00991	0.00000	−1.00	−3.11
	H7-N6	0.00763	0.02552	0.00536	−0.00434	0.00102	−0.81	−1.36
**NH_3_@C_6_N_6_**	N8-N1	0.00773	0.02558	0.00573	−0.00506	0.00067	−0.88	−1.59
	H7-N2	0.01365	0.04227	0.00991	−0.00925	0.00066	−0.93	−2.90
	H7-N3	0.00867	0.02856	0.00627	−0.00540	0.00087	−0.86	−1.69
	N8-N4	0.00357	0.01222	0.00261	−0.00217	0.00044	−0.83	−0.68
	H9-N5	0.00874	0.02871	0.00631	−0.00545	0.00087	−0.86	−1.71
	H9-N6	0.01360	0.04224	0.00989	−0.00922	0.00067	−0.93	−2.89
**PH_3_@C_6_N_6_**	P7-N1	0.00687	0.02064	0.00445	−0.00374	0.00071	−0.84	−1.17
	H8-N2	0.00562	0.01924	0.00390	−0.00300	0.00091	−0.77	−0.94
	H8-N3	0.00673	0.02215	0.00456	−0.00358	0.00098	−0.79	−1.12
	H8-N4	0.00730	0.02305	0.00482	−0.00388	0.00094	−0.80	−1.22
	H8-N5	0.00601	0.02070	0.00416	−0.00315	0.00101	−0.76	−0.99
	H7-N6	0.00625	0.02024	0.00435	−0.00365	0.00071	−0.84	−1.14

**Table 3 nanomaterials-13-01121-t003:** The energies of the HOMO, LUMO and HOMO–LUMO energy gap (EH–L gap) of analyte@C_6_N_6_ complexes.

Analyte@C_6_N_6_	HOMO (eV)	LUMO (eV)	Gap
**C_6_N_6_**	−7.16	−3.30	3.86
**HCN@C_6_N_6_**	−7.43	−3.41	4.02
**H_2_S@C_6_N_6_**	−6.07	−3.31	2.76
**NH_3_@C_6_N_6_**	−5.85	−3.27	2.58
**PH_3_@C_6_N_6_**	−6.62	−3.27	3.35

**Table 4 nanomaterials-13-01121-t004:** Comparison of adsorption energy of H-containing industrial pollutants over C_6_N_6_ with already reported values of different analytes over C_6_N_6_ and other COFs.

S.No	Complexes	E_ads_ (kcal/mol)	Ref	S.No	Complexes	E_ads_ (kcal/mol)	Ref
1	NH_3_@CTF-FUM	−5.27	[5]	8	HCN@CTF-0	−5.99	[86]
2	H_2_S@CTF-FUM	−3.79		9	H_2_S@CTF-0	−5.46	
3	PH_3_@CTF-FUM	−2.43		10	NH_3_@CTF-0	−6.65	
4	HCN@C_2_N	−15.24		HCN@C_6_N_6_		−16.46	current work
5	NH_3_@C_2_N	−10.68		PH_3_@C_6_N_6_		−13.64	
6	H_2_S@C_2_N	−8.54		NH_3_@C_6_N_6_		−12.27	
7	PH_3_@C_2_N	−6.91		PH_3_@C_6_N_6_		−9.36	

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
