# Peer review of "Covalent Triazine Framework C6N6 as an Electrochemical Sensor for Hydrogen-Containing Industrial Pollutants. A DFT Study"

_nanomaterials, 2023, doi:10.3390/nano13061121_

Round 1

Reviewer 1 Report

This is an interesting manuscript that can be accepted for publication. Minor English corrections are needed. The introduction is very good and methods used for numerical investigations also. The abstract includes too many acronyms.  

Author Response

Reviewer 1

Comments and Suggestions for Authors

This is an interesting manuscript that can be accepted for publication. Minor English corrections are needed. The introduction is very good and methods used for numerical investigations also. The abstract includes too many acronyms.  

Response: We thank the reviewer for the positive comments and acceptance of our manuscript for publication in Nanomaterials. The entire manuscript is proofread carefully to remove English grammar and typographical mistakes. In addition, a full description of each acronym is added in the abstract of the manuscript.

Reviewer 2 Report

The authors have reported a detailed DFT study using the covalent triazine framework C6N6 as an electrochemical sensor for hydrogen-containing industrial pollutants. The authors have clearly presented the non-covalent interaction of different industrial pollutants (HCN, H2S, NH3 and PH3) with the C6N6 through the SAPT0, QTAIM and NCI analysis. EDD, NBO and FMO further investigate the electronic properties of analytes/C6N6 complexes. The manuscript has significant and valuable scientific information to be considered for publication in Nanomaterials. However, the authors need to respond to some questions before.

1.   In the Introduction, please explain more benefits of triazine compared with another covalent organic framework.

2.   Please explain how high nitrogen-containing traizine is more efficient to capture hydrogen-containing target molecules.

3.   In the SAPTO analysis, the four factors including induction, exchange, electrostatic and dispersion plays an important role. However, in HCN, H2S, NH3, the Eelec factor is predominant, but why the electrostatic interaction is low for PH3?

4.    Please explain why the Einduction factor is low in all cases.

5.  The authors reported that the out of three hydrogen atoms in PH3, only one H atom is oriented to C6H6 cavity. However, the structure of NH3 is very similar to PH3 as trigonal pyramidal. If so, why does the interaction energy differ a lot for NH3 and PH3?

6. If two types of target molecules are present in the same media, can we predict its interaction using DFT?

7. Please revise the grammatical, subscript, and spacing errors in the manuscript.

Author Response

Reviewer 2.

Comments and Suggestions for Authors

The authors have reported a detailed DFT study using the covalent triazine framework C6N6 as an electrochemical sensor for hydrogen-containing industrial pollutants. The authors have clearly presented the non-covalent interaction of different industrial pollutants (HCN, H2S, NH3 and PH3) with the C6N6 through the SAPT0, QTAIM and NCI analysis. EDD, NBO and FMO further investigate the electronic properties of analytes/C6N6 complexes. The manuscript has significant and valuable scientific information to be considered for publication in Nanomaterials. However, the authors need to respond to some questions before.

  1. In the Introduction, please explain more benefits of triazine compared with another covalent organic framework.

Response: We thank the reviewer for the constructive suggestions to improve our manuscript. We have added comparison of CTFs with other COFs as given below. The same is added as the 2nd  last paragraph of “introduction” section.

The previous reports show that covalent triazine frameworks (CTFs) have been explored as better sensor compared to other COFs, due to controlled porosity, high density nitrogen cavity, high surface area, chemical as well as thermal stability, reproducible sensing response and tunable electronic properties [49][50]. These characteristics provide suitable condition for trapping and detection of analytes and make CTFs more potential sensor surfaces compared to other COFs [51–53].

  1. Please explain how high nitrogen-containing traizine is more efficient to capture hydrogen-containing target molecules.

Response: We selected recently designed covalent triazine framework (C6N6), which consist of six nitrogens in its cavity. High density nitrogen containing C6N6 unit also keeps high surface area with cavity size of 5.46 Å which can efficiently interact with H-containing industrial pollutants like HCN, H2S, NH3 and PH3.

The above given statement is included in the last paragraph of “introduction” section.

  1.  In the SAPTO analysis, the four factors including induction, exchange, electrostatic and dispersion plays an important role. However, in HCN, H2S, NH3, the Eelecfactor is predominant, but why the electrostatic interaction is low for PH3?

Response: The H-atoms of HCN, H2S and NH3 are comparatively more polar compared to the H-atoms of PH3. Due to which HCN, H2S and NH3 are stabilized through Eelst factor, while PH3 is stabilized through dispersion interactions rather than electrostatic.

  1.  Please explain why the Einduction factor is low in all cases.

Response:  In SAPT0, interaction between occupied orbitals of one fragment with virtual orbitals of other fragments are accounted by ΔEind. In our studied complexes, the major contributing factors towards stabilization of complexes are Eelst because interacting analytes are mostly polar. The low value of Einduction in all studied complexes may be due to slight interaction between occupied and virtual orbitals of interacting systems.

  1. The authors reported that the out of three hydrogen atoms in PH3, only one H atom is oriented to C6H6cavity. However, the structure of NH3 is very similar to PH3 as trigonal pyramidal. If so, why does the interaction energy differ a lot for NH3 and PH3?

Response: In case of NH3, inter-atomic distance of 1.62 Å is observed between two H-atoms attached to N-atom of NH3. Whereas in case of PH3, H-atoms are at the distance of 2.07 Å. Due to larger inter-atomic distance between H-atoms of PH3, only H-atom found in interaction with N-atoms of C6N6. Whereas, in case of NH3, two H-atoms interacted with ­C6N6 due to shorter inter-atomic distance between H-atoms of NH3

  1. If two types of target molecules are present in the same media, can we predict its interaction using DFT?

Response: We say sorry that we are not clear that what the reviewer is trying to ask. According to our understanding, if two types of molecules are present in same media, the one which will interact strongly is preferably get stable complex (the one with highest interaction energy). We have not performed cooperative adsorption in current study. We are planning to do such types of study in future.

  1. Please revise the grammatical, subscript, and spacing errors in the manuscript.

Response: We thank the reviewer for mentioning it. We have corrected all typos and grammatical errors in the revised manuscript.

Reviewer 3 Report

This article is devoted to a theoretical study of the C6N6 triazine backbone. The article presents various docking of this carcase with hydrogen-containing pollutants. The article has a sufficient volume and amount of data received for the Journal. There are some points that could be improved:

1. A clearer statement of purpose is desirable.

2. It is advisable to correct "B3LPY/6-31G" to "B3LYP/6-31G" where necessary.

3. It is advisable to double-check the text for grammatical and stylistic errors.

4. It is desirable to formulate the rationale for choosing this particular calculation method, functional and basis for this study.

5. In the "Results and discussion" part, there is little comparison with data from the literature. It is desirable to add.

6. Please cite: 10.1007/s11696-020-01220-3.

7. Conclusions can be made more concise.

Author Response

Reviewer 3

Comments and Suggestions for Authors

This article is devoted to a theoretical study of the C6N6 triazine backbone. The article presents various docking of this carcase with hydrogen-containing pollutants. The article has a sufficient volume and amount of data received for the Journal. There are some points that could be improved:

  1. A clearer statement of purpose is desirable.

Response: Industrial pollutants pose a serious threat to the ecosystem. Hence, it is a need of time to search new efficient sensor for the detection of these pollutants and which can overcome the existing limitation of existing 2D-materials like low porosity, less surface area, a smaller number of active site, poor selectivity as well as sensitivity issues and poor recyclability.

  1. It is advisable to correct "B3LPY/6-31G" to "B3LYP/6-31G" where necessary.

Response: we have corrected the mentioned mistake in methodology section.

  1. It is advisable to double-check the text for grammatical and stylistic errors.

Response: We have made all the corrections in the revised manuscript.

  1. It is desirable to formulate the rationale for choosing this particular calculation method, functional and basis for this study.

We thank reviewer for his critical comments. We carried out our DFT study on ꞶB97XD functional based on previously reported benchmark studies. The Ꞷb97XD functional is adopted in the current study because it is a range-separated functional which can effectively capture the non-localized intermolecular interactions [58]. In Ꞷb97XD, symbol “Ꞷ” long range correction [59]. Additionally, Ꞷb97XD functional also contains Grimme’s D2 dispersion factor to efficiently study the van der Waal (vdWs) interactions [59].

  1. In the “Results and discussion” part, there is little comparison with data from the literature. It is desirable to add.

Response: We compared the adsorption energies of our studied system with those available in the literature for different COFs. The adsorption energies of H-containing analytes are comparable or somewhat higher than already reported values on other surfaces. The adsorption energy of -12.27 kcal/mol is observed for NH3@C6N6 complex. Whereas in literature, adsorption energies of -5.27 kcal/mol, -10.68 kcal/mol and -6.65 kcal/mol are observed for NH3@CTF-FUM, NH3@C2N and NH3@CTF-0 complexes, respectively. In addition, interaction energies of -15.24 kcal/mol, -3.79 kcal/mol and -2.34 kcal/mol are observed for HCN@CTF-FUM, H2S@CTF-FUM and PH3@CTF-FUM, respectively. The adsorption energies of rest of the complexes given in Table 4 reveal that C6N6 can as better surfaces for electrochemical sensing of H-containing analytes

Table 4: Comparison of adsorption energy of H-containing industrial pollutants over C6N6 with already reported values of different analytes over C6N6 and other COFs.

S.No

Complexes

Eads (kcal/mol)

Ref

S.No

Complexes

Eads (kcal/mol)

Ref

1

NH3@CTF-FUM

−5.27

[5]

8

HCN@CTF-0

-5.99

[86]

2

H2S@CTF-FUM

−3.79

9

H2S@CTF-0

-5.46

3

PH3@CTF-FUM

−2.43

10

NH3@CTF-0

-6.65

4

HCN@C2N

-15.24

[63]

HCN@C6N6

-16.46

current work

5

NH3@C2N

−10.68

PH3@C6N6

-13.64

6

H2S@C2N

-8.54

NH3@C6N6

-12.27

7

PH3@C2N

-6.91

PH3@C6N6

-9.36

The above-mentioned Table is now added in the revised manuscript.

  1. Please cite: 10.1007/s11696-020-01220-3.

Response: We thank the reviewer for providing us valuable literature. The given reference is added in computational methodology [see ref.65]

  1. Conclusions can be made more concise.

Response: As per reviewer suggestions, we have removed many statements from conclusions to make it more concise. Now conclusions contain only significant information related to results of each section.